# Three-Year Analysis of Adjuvant Therapy in Postoperative Melanoma including Acral and Mucosal Subtypes

**DOI:** 10.3390/cancers16152755

**Published:** 2024-08-02

**Authors:** Yusuke Muto, Yumi Kambayashi, Hiroshi Kato, Satoru Mizuhashi, Takamichi Ito, Takeo Maekawa, Shoichiro Ishizuki, Hiroshi Uchi, Shigeto Matsushita, Yuki Yamamoto, Koji Yoshino, Yasuhiro Fujisawa, Ryo Amagai, Kentaro Ohuchi, Akira Hashimoto, Satoshi Fukushima, Yoshihide Asano, Taku Fujimura

**Affiliations:** 1Department of Dermatology, Tohoku University Graduate School of Medicine, Sendai 980-8574, Japan; yusuk0610@derma.med.tohoku.ac.jp (Y.M.); kambayashi@derma.med.tohoku.ac.jp (Y.K.); amagai@derma.med.tohoku.ac.jp (R.A.); r2v7ib@bma.biglobe.ne.jp (K.O.); hashi-a@derma.med.tohoku.ac.jp (A.H.); yasano@derma.med.tohoku.ac.jp (Y.A.); 2Department of Geriatric and Environmental Dermatology, Nagoya City University Graduate School of Medical Sciences, Nagoya 467-8601, Japan; hkato43@gmail.com; 3Department of Dermatology and Plastic Surgery, Faculty of Life Sciences, Kumamoto University, Kumamoto 860-8556, Japan; mizuhashi.satoru.kum@gmail.com (S.M.); s_fukushima@kumamoto-u.ac.jp (S.F.); 4Department of Dermatology, Graduate School of Medical Science, Kyushu University, Fukuoka 812-8582, Japan; ito.takamichi.657@m.kyushu-u.ac.jp; 5Department of Dermatology, Jichi Medical University Saitama Medical Center, Saitama 330-8503, Japan; maekawat@jichi.ac.jp; 6Department of Dermatology, Faculty of University of Tsukuba, Tsukuba 305-8575, Japan; syoichiro200711668@gmail.com; 7Department of Dermato-Oncology, NHO Kyushu Cancer Center, Fukuoka 811-1395, Japan; uchih@icloud.com; 8Department of Dermato-Oncology/Dermatology, NHO Kagoshima Medical Center, Kagoshima 892-0853, Japan; shigeto0302@gmail.com; 9Department of Dermatology, Wakayama Medical University, Wakayama 641-0012, Japan; yukiy@wakayama-med.ac.jp; 10Department of Dermato-Oncology/Dermatology, Cancer Institute Hospital of Japanese Foundation for Cancer Research, Tokyo 135-8550, Japan; koyoshin24@gmail.com; 11Department of Dermatology, Ehime University, Matsuyama 791-0295, Japan; fujisawa.yasuhiro.fp@ehime-u.ac.jp

**Keywords:** acral, mucosal, anti-PD-1 Abs, BRAF plus MEK inhibitors, adjuvant therapy

## Abstract

**Simple Summary:**

This study aimed to assess the 3-year time to relapse (TTR) and overall survival (OS) of melanoma, including acral and mucosal subtypes, treated with anti-PD-1 antibody (Ab) or the combination of dabrafenib and trametinib. The 3-year TTR of the acral and mucosal types was 28.1% and 38.5%, respectively. The acral subtype and TT were detected as important prognostic factors. In the 3-year OS, only tumor ulceration was associated with the OS in both univariate and multiple analyses. There was no significant difference in baseline or treatment-related factors of the mucosal type (*p* > 0.05). This study suggests that both acral and mucosal types in the adjuvant setting are less effective than non-acral cutaneous melanoma at the 3-year TTR.

**Abstract:**

Background: Adjuvant therapy has improved the clinical prognosis for postoperative melanoma patients. However, the long-term efficacy of this therapy on the melanoma acral and mucosal subtypes has not been fully evaluated in previous trials. This study assessed the 3-year recurrence-free survival and overall survival of patients with melanoma, including the acral and mucosal subtypes, treated with anti-PD-1 antibody (Ab) or with the combination of the BRAF and MEK inhibitors dabrafenib and trametinib. Methods: We retrospectively analyzed both the 3-year time to relapse (TTR) and overall survival (OS) of 120 patients treated with anti-PD-1 antibody (Ab), or with the combination of dabrafenib and trametinib. Results: The overall median TTR was 18.4 months, with a range of 0.69 to 36 months. The 3-year TTR of the acral and mucosal types was 28.1% and 38.5%, respectively. Baseline tumor thickness (TT) and acral type were associated with the TTR in subgroup analysis. Moreover, we classified 104 acral and non-acral cutaneous patients into the anti-PD-1 Abs or dabrafenib plus trametinib combined therapies cohort in multiple analyses. The acral subtype and TT were detected as important prognostic factors. In the 3-year OS, only tumor ulceration was associated with the OS in both univariate and multiple analyses. There was no significant difference in baseline or treatment-related factors of the mucosal type (*p* > 0.05). Conclusion: This study suggests that adjuvant therapy is more effective with non-acral cutaneous melanoma than either the acral or mucosal types at the 3-year TTR endpoint.

## 1. Introduction

Adjuvant therapy, including immune checkpoint inhibitors targeting programmed cell death 1 (PD-1) or cytotoxic T-lymphocyte-associated antigen 4 (CTLA-4) proteins, and targeted therapy with BRAF plus MEK inhibitors, has improved the clinical prognosis for postoperative melanoma patients at high risk of recurrence. In the CheckMate 238 trial, the nivolumab group treated with 3 mg/kg once every 2 weeks for 1 year demonstrated a 4-year recurrence-free survival (RFS) rate of 51.7% and a 4-year overall survival (OS) of 77.9%, compared to 76.6% for ipilimumab in resected stage III-IV melanoma [1]. The EORTC 1325-MG/KEYNOTE-054 study showed that pembrolizumab treatment with 200 mg every 3 weeks for up to 18 doses or until disease recurrence or unacceptable toxicity resulted in a 3.5-year RFS rate of 59.8–61.4%, compared to 41.4–44.1% in the placebo group [2]. In the KEYNOTE-716 Study, pembrolizumab achieved an 84.4% distant metastasis-free survival compared to 74.7% in the placebo group for Stage IIB or IIC disease [3]. The COMBI-AD trial demonstrated that combined therapy with dabrafenib plus trametinib yielded a 5-year RFS rate of 52%, compared to 36% in the placebo group for resected stage III melanoma [4]. However, as the incidence of mucosal and acral melanoma is higher in Asian cohorts, including Japanese populations (mucosal, 20–30%; acral type, 42–65% of all melanoma types), than in Caucasians (mucosal, 1–2%; acral melanoma, 1–1.5% of all melanoma cases) [5], a small number of acral and mucosal melanoma cases have been included in clinical trials.

Thus, this study aims to analyze the 3-year time to relapse (TTR) and OS in 120 melanoma cases with stage IIC/III/IV in the adjuvant setting, treated with anti-PD-1 Abs or the combination of dabrafenib plus trametinib. The purpose is to evaluate the long-term effectiveness of adjuvant anti-PD-1 Abs or dabrafenib plus trametinib in a Japanese cohort, including more acral and mucosal patients.

## 2. Materials and Methods

### 2.1. Ethics

The protocol was approved by the ethics committee of Tohoku University Graduate School of Medicine, Sendai, Japan (2020-1-811), and by the ethics committees of each participating institution.

### 2.2. Patients

From January 2019 to September 2020, we retrospectively identified 120 patients who had been treated with anti-PD-1 (nivolumab or pembrolizumab) or BRAF plus MEK inhibitors (dabrafenib plus trametinib) in the adjuvant setting, as previously reported [6].

The patients had histologically confirmed cutaneous, acral, or mucosal melanoma and were categorized as either stage IIC or stage IIIA-C or had resectable IV disease according to the eighth edition of the Cancer Staging Manual of the American Joint Committee on Cancer (AJCC-8th). Additionally, histologic subtypes of cutaneous melanoma were evaluated using both Clark’s histological classification and the World Health Organization (WHO) classification, which categorizes them based on the histopathologic degree of cumulative solar damage (CSD) of the surrounding skin. In stage IV disease, adjuvant therapy was initiated after complete resection of distant or recurrent lesions.

### 2.3. Procedure

Patients with BRAF-wild type melanoma were treated with anti-PD-1 inhibitors (pembrolizumab or nivolumab). Nivolumab was administered intravenously at a dose of 240 mg every two weeks or 480 mg every four weeks, while pembrolizumab was given at a dose of 200 mg every three weeks for 12 months. Patients with BRAF-mutant melanoma received treatment with either anti-PD-1 inhibitors or dabrafenib plus trametinib combined therapy. Adjuvant therapy was continued for 12 months or until disease recurrence or unacceptable toxicity occurred. Withdrawal of the agents was approved for severe adverse events by the decision of the physician.

### 2.4. Assessment

The primary endpoint was the 3-year time-to-relapse (TTR) rate, defined as the time from adjuvant therapy induction to disease recurrence or the last date of examination. The secondary endpoint was the 3-year overall survival (OS), defined as the time from adjuvant therapy induction to the patient’s death from any cause. All disease recurrence analyses were based on investigator assessment. Periodic checkups and imaging, using computed tomography (CT), magnetic resonance imaging (MRI), or both, were performed for the 3-year follow-up.

### 2.5. Statistical Analysis

The TTR rate was estimated for each group using the Kaplan–Meier method. The log-rank test was used to compare survival between groups. Hazard ratios (HRs) and 95% confidence intervals (95% CIs) were estimated using Cox’s proportional hazards model in the univariate analysis. Cox’s proportional hazards models were also utilized for multivariate analyses (estimate age, melanoma subtype, BRAF status, tumor thickness or ulceration, sex, and lymph node dissection). The significance level for all tests was set at a two-sided α = 0.05. The significance level for the log-rank test was also set at a two-sided α = 0.05. All statistical analyses were performed with JMP ver.16 (SAS Institute Inc., Cary, NC, USA).

## 3. Results

### 3.1. Demographic Data

The patients’ demographic data have been described previously [6].

Briefly, in the anti-PD-1 inhibitor group, 1 (1%) patient had stage IIC, 3 (3%) had stage IIIA, 12 (13%) had stage IIIB, 46 (52%) had stage IIIC, and 3 (3%) had stage IIID, while in the dabrafenib plus trametinib combined therapies group, 1 (3%) patient had stage IIIA, 6 (20%) had stage IIIB, 17 (57%) had stage IIIC, and 3 (10%) had stage IIID. Before adjuvant therapy, 26 patients with cutaneous and acral melanoma at stage III underwent complete lymph node dissection (CLND), while 65 patients did not.

### 3.2. Three-Year TTR in Subtypes of Melanoma

The median TTR for the 120 patients was 18.4 months, ranging from 0.69 to 36 months. The 3-year TTR rate was 35.8% (43/120 cases). In subtype-specific analysis, the 3-year TTR rate was 26.2% (11/42 cases) for acral melanoma, 44% (11/25 cases) for high-CSD, 43.2% (16/37 cases) for low-CSD, 33.3% (4/12 cases) for mucosal melanoma, and 50% (2/4 cases) for unknown primary sites. Non-acral cutaneous type (low- and high-CSD types) treated with anti-PD-1 Abs or dabrafenib plus trametinib combined therapy showed a significantly better TTR compared to acral melanoma (HR, 0.56; 95% CI: 0.34–0.92; *p* = 0.021, Figure 1a). Additionally, significant differences were observed between acral and non-acral melanoma in the anti-PD1 Abs cohort (HR, 0.52; 95% CI, 0.30–0.89; *p* = 0.016, Figure 1b). Mucosal melanoma did not exhibit a poorer prognosis compared to non-acral melanoma, consistent with the previous 2-year analysis (*p* = 0.46).

We divided both cutaneous and acral melanoma patients (*n* = 104) into two groups based on baseline or treatment-related factors. Patients with a thin tumor thickness (TT) < 4 mm showed a significantly better TTR than those with a thick TT ≥ 4 mm (median TT value: 4 mm; HR, 0.39; 95% CI, 0.23–0.68; *p* = 0.0006, Figure 1c). There was no significant difference between tumor ulcer-positive and -negative cases (*p* = 0.18, Figure 1d). BRAF-positive cases had a better prognosis than BRAF-negative cases (HR, 0.47; 95% CI, 0.26–0.85; *p* = 0.01, Figure 1e). CLND was not associated with TTR in stage III melanoma (*p* = 0.23, Figure 1f).

### 3.3. Three-Year OS in Subtypes of Melanoma

The median overall survival (OS) was 36.0 months (range, 1.5–36 months). The 3-year OS rates for acral, high-CSD, low-CSD, mucosal, and unknown primary types were 47.6% (20/42 cases), 60% (15/25 cases), 64.9% (24/37 cases), 33.3% (4/12 cases), and 50% (2/4 cases), respectively. The 3-year OS based on the AJCC-8th staging was 75% for stage IIIA, 82.9% for stage IIIB, 54.1% for stage IIIC, and 75% for stage IIID disease in both acral and non-acral cutaneous melanomas. We analyzed the OS between acral and non-acral melanomas, and there were no significant differences (*p* > 0.05, Figure 2a,b). The mucosal type did not exhibit a poorer prognosis compared to non-acral melanoma in terms of OS (*p* > 0.05). Regarding baseline or treatment-related factors, patients with tumor ulceration had a significantly better prognosis in terms of OS than those without tumor ulceration (HR, 0.39; 95% CI: 0.17–0.85; *p* = 0.013, Figure 2d). There were no significant differences in median TT, tumor ulceration, BRAF status, and CLND (*p* > 0.05, Figure 2c,e,f).

### 3.4. Multivariate Analysis of the Prognostic Factors in Both TTR and OS

Among 104 acral and non-acral cutaneous melanoma cases, we evaluated histological characteristics (acral type, TT, ulceration, and BRAF status), baseline characteristics (gender and age), and treatment options for multivariate analysis to identify relevant prognostic factors for TTR or OS. Moreover, we classified 104 patients into the anti-PD-1 Abs or dabrafenib plus trametinib combined therapies cohort.

In the anti-PD-1 Abs cohort, non-acral cutaneous type (HR, 0.33; 95% CI, 0.17–0.64; *p* = 0.0010) and median tumor thickness (HR, 0.36; 95% CI, 0.19–0.69; *p* = 0.0022) showed statistically significant differences in TTR (Table 1), while in the combined therapies cohort, there were no significant factors (*p* > 0.05, Appendix A). Only tumor ulceration was associated with OS in the anti-PD-1 Abs’ cohort (HR, 0.30; 95% CI, 0.095–0.97; *p* = 0.0044, Table 2). There was no significant difference in the dabrafenib plus trametinib combined therapies cohort (*p* > 0.05, Appendix A).

## 4. Discussion

Acral melanoma exhibits a higher incidence in Asian countries, including Japan, compared to European countries and the U.S. (incidence rate: 1.0–1.5% vs. 42–65%) [5]. Acral melanoma is characterized by higher rates of BRAF-wild type, lower tumor mutation burden, and a lack of ultraviolet (UV)-related mutational signatures compared to non-acral cutaneous melanoma [7]. The numbers, localization, and phenotypes of tumor-infiltrating lymphocytes (TILs) in the tumor microenvironment are well-known indicators reflecting response to immunotherapy [8,9]. CD8+ T cell infiltration has been identified in multiple studies as an important prognostic factor for clinical outcomes and predictive value for response to immunotherapy [10]. Additionally, acral melanoma was characterized by fewer NK cells and an almost complete absence of γδ T cells in the TILs [11]. Since NK cells express cytotoxic proteins commonly found in CD8+ T cells that directly kill target cells [12], acral melanoma may have an immunosuppressed environment compared to non-acral cutaneous melanoma. These studies suggest that anti-PD-1 Abs are less effective in acral melanoma. In fact, several studies, including non-Asian cases, confirm that anti-PD-1 Abs are less effective in acral melanoma, even in adjuvant therapy [13,14,15,16].

In our 3-year analysis, we demonstrated that acral melanoma had a poorer prognosis compared to non-acral cutaneous melanoma in terms of TTR in the adjuvant setting (*p* < 0.005, log-rank analysis). Significant differences were observed in TT and BRAF status at the 3-year TTR. In our cohort, most patients with BRAF-mutated melanoma opted for combination therapy with BRAF plus MEK inhibitors. In the Asian population, immune checkpoint inhibitors have shown lower responsiveness to advanced melanoma in several reports compared to clinical trials from Caucasians. Therefore, dermatologists or oncologists in Japan tend to administer BRAF + MEK inhibitors to BRAF-mutated cases [17,18]. Other histological characteristics (mucosal type, tumor ulceration), baseline characteristics (sex and age) did not show significant differences in TTR (*p* > 0.05). In multivariate analysis, acral melanoma and TT showed significant differences among 104 cutaneous and acral melanoma cases. We demonstrated that acral melanoma was one of the independent prognostic factors at 3-year TTR, as well as at the 2-year TTR. Since patients treated with a combination of BRAF + MEK inhibitors had better TTR than those treated with anti-PD-1 inhibitors, non-acral types with higher BRAF mutation were preferred for the combination therapy, while acral melanoma with lower BRAF mutation received anti-PD-1 inhibitors.

From 1948 patients with cutaneous melanoma, the 5-year melanoma-specific survival (MSS) for stage IIIA-C was 89%, 81%, and 64%, respectively [19]. In the Japanese Melanoma Study (JMS) using AJCC-8th criteria, the 3-year disease-specific survival (DSS) was 92.3% for stage IIIA, 82.5% for stage IIIB, 67.1% for stage IIIC, and 38.5% for stage IIID disease in Asian populations [20]. Notably, these data included 3097 patients from 2005 to 2017. Our 3-year OS results were 75% for stage IIIA, 82.9% for stage IIIB, 54.1% for stage IIIC, and 75% for stage IIID disease in both acral and non-acral cutaneous melanoma. Unexpectedly, postoperative adjuvant therapy was not shown to be superior to previous literature. We suspect that the difference in sample size and melanoma subtype in our cohort had an impact on the results. Additionally, our data included patients who died from any cause, so we could not accurately assess the effectiveness of adjuvant therapy in stage III. In our cohort, patients treated with CLND tended to show better OS compared to those without CLND in the univariate analysis (*p* = 0.051).

## 5. Conclusions

This study suggests that both the acral and mucosal types in the adjuvant setting are less effective than adjuvant therapy of non-acral cutaneous melanoma at the 3-year TTR. We retrospectively analyzed the 3-year TTR and OS in real-world melanoma patients in the adjuvant setting; however, this was not a large cohort study in the Asian population. It is difficult to evaluate the efficacy of adjuvant anti-PD-1 or dabrafenib plus trametinib combination therapy on OS compared with conventional adjuvant therapy or observation. As there are no prospective trials of the benefits and harms of lymph node dissection, it is not possible to make a definitive statement about its role in melanoma treatment. However, in this retrospective study, lymph node dissection tends to improve OS, although the small number of cases prevents a statistically significant difference. Further investigation was needed with a large sample size and low bias.

## Figures and Tables

**Figure 1 cancers-16-02755-f001:**
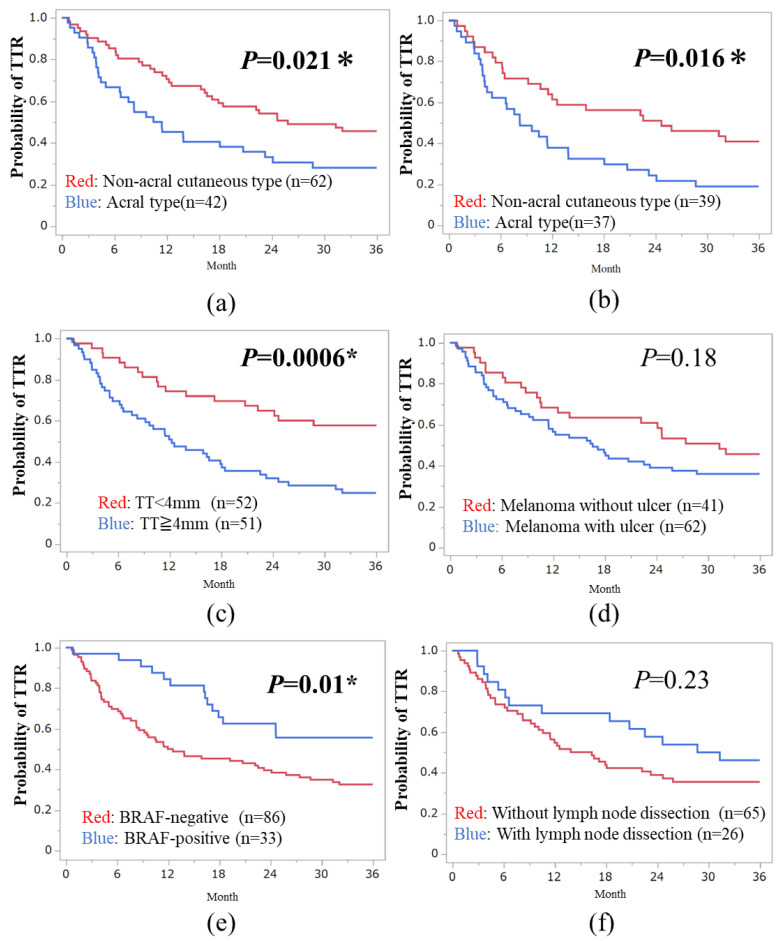
Assessment of Recurrence-Free Survival in Post-Operative Melanoma. (**a**) Comparison between acral and non-acral cutaneous types treated with anti-PD-1 or BRAF + MEK inhibitors. (**b**) Comparison between acral and non-acral cutaneous types treated with anti-PD-1 inhibitor. (**c**) Time to relapse (TTR) according to median tumor thickness. (**d**) TTR according to tumor ulceration. (**e**) TTR comparison between BRAF-positive and -negative cases. (**f**) TTR comparison between cases with and without complete lymph node dissection. * *p* < 0.05.

**Figure 2 cancers-16-02755-f002:**
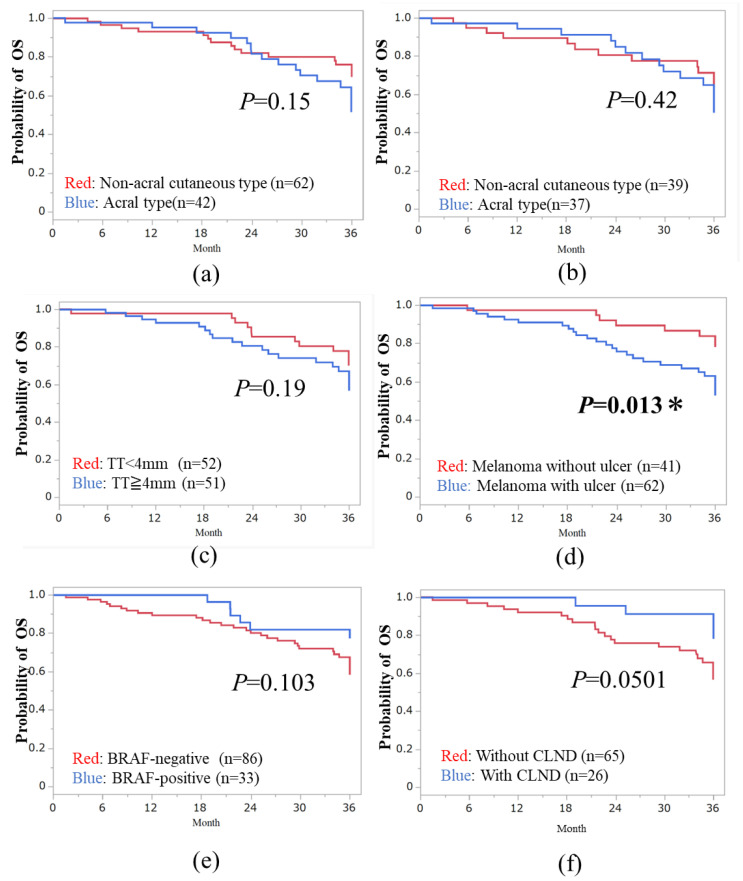
Assessment of Overall Survival in Post-Operative Melanoma. (**a**) Comparison between acral and non-acral cutaneous types treated with anti-PD-1 or BRAF + MEK inhibitors. (**b**) Comparison between acral and non-acral cutaneous types treated with anti-PD-1 inhibitor. (**c**) Overall survival (OS) according to median tumor thickness. (**d**) OS according to tumor ulceration. (**e**) OS comparison between BRAF-positive and -negative cases. (**f**) OS comparison between cases with and without complete lymph node dissection. * *p* < 0.05.

**Table 1 cancers-16-02755-t001:** Multivariate Cox proportional hazards model analysis of prognostic factors associated with the TTR in anti-PD-1 Abs’ cohort (*n* = 76). * *p* < 0.05.

Parameters	Hazard Ratio (95% CI)	*p* Value
**Histological characteristics**		
Non-acral cutaneous vs. acral types	0.28 (0.14–0.59)	0.0007 *
Tumor thickness	0.32 (0.16–0.66)	0.0020 *
Tumor ulceration	0.83 (0.40–1.7)	0.61
BRAF wild vs. mutated type	0.44 (0.12–1.6)	0.21
**Baseline characteristics**		
Sex	0.65 (0.34–1.3)	0.2
With or Without lymph node dissection	0.77 (0.40–1.5)	0.45

**Table 2 cancers-16-02755-t002:** Multivariate Cox proportional hazards model analysis of prognostic factors associated with the OS in anti-PD-1 Abs cohort (*n* = 76). * *p* < 0.05.

Parameters	Hazard Ratio (95% CI)	*p* Value
**Histological characteristics**		
Non-acral cutaneous vs. acral types	0.95 (0.38–2.33)	0.9
Tumor thickness	0.83 (0.32–2.2)	0.71
Tumor ulceration	0.30 (0.095–0.97)	0.044 *
BRAF wild vs. mutated type	NA	0.99
**Baseline characteristics**		
Sex	0.72 (0.28–1.9)	0.5
With or Without lymph node dissection	0.77 (0.40–1.5)	0.45

## Data Availability

The data that support the findings of this study are available on request from the corresponding author, T.F. The data are not publicly available due to containing information that could compromise the privacy of research participants.

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
