# Peer review of "Three-Year Analysis of Adjuvant Therapy in Postoperative Melanoma including Acral and Mucosal Subtypes"

_cancers, 2024, doi:10.3390/cancers16152755_

Round 1

Reviewer 1 Report

Comments and Suggestions for Authors

This article outlines the effects of adjuvant immunotherapy in acral and mucosal subtypes. The paper's conclusion is generally well-accepted by the field.  

1. Some moderate English language editing is required.  For example, therapy is effective, but not the type of melanoma. 

2. Can conclusions be made regarding mutations other than BRAF for this study (e.g., NRAS)?  

3. What do the authors believe is the role of lymph node dissection in the acral subtype? 

4. Recommend using statistical testing for multiple comparisons.  

Comments on the Quality of English Language

I recommend English editing as described above.  

Author Response

Comments and Suggestions for Authors

This article outlines the effects of adjuvant immunotherapy in acral and mucosal subtypes. The paper's conclusion is generally well-accepted by the field.  

  1. Some moderate English language editing is required.  For example, therapy is effective, but not the type of melanoma. 

As you and reviewer 3 suggested, we corrected English language.

  1. Can conclusions be made regarding mutations other than BRAF for this study (e.g., NRAS)?  

Only BRAF inhibitors are currently being tested in this analysis for BRAF gene mutations, as there is currently clear evidence of their therapeutic efficacy by previous clinical studies. Analysis of other mutations would require large-scale testing, such as genomic gene panel testing, which is not the purpose of this study.

  1. What do the authors believe is the role of lymph node dissection in the acral subtype? 

As there are no prospective trials of the benefits and harms of lymph node dissection, it is not possible to make a definitive statement about its role in melanoma treatment. However, in this retrospective study, lymph node dissection tends to improve OS, although the small number of cases prevents a statistically significant difference. This will be added to the text.

  1. Recommend using statistical testing for multiple comparisons.  

We have already used Cox proportional hazard model for multivariate analyses for OS and time to relapse (Age, melanoma subtype, BRAF status, tumor thickness or ulceration, sex, and lymph node dissection) in this study (L114- 115).

Reviewer 2 Report

Comments and Suggestions for Authors

This manus is interesting since  in clinic it is needed to clarify the inpact in real life of the adjuvant setting.

I suggest that the Relapse free survival would be define better , It seems it is a  Time to relapse (TTR) to me, The relapse free cases are defined as "status" free of relapse at the time of censoring (is censoring a define datum for every patients in this case when)

It is missing a table with the presentation of the  study with information on the number of patients  gender  clinical data  etc.

The cohort presented is quite heterogeneous with small subgroups that show the difficult to have reliable differences and sustainable clinical use.

Author Response

I suggest that the Relapse free survival would be define better, It seems it is a Time to relapse (TTR) to me, The relapse free cases are defined as "status" free of relapse at the time of censoring (is censoring a define datum for every patients in this case when)

Thank you for your suggestion. We changed the Relapse free survival (RFS) into Time to relapse (TTR) in the manuscript and figures.

It is missing a table with the presentation of the study with information on the number of patients gender clinical data etc.

This study was published in Acta DV and Brit J Dermatol with results for the first and second year respectively. Therefore, these epidemiological data were already available in the first year, so the data are available in this paper in the form of citations (ref.6, 7).

The cohort presented is quite heterogeneous with small subgroups that show the difficult to have reliable differences and sustainable clinical use.

As you pointed out, this point is listed as a limitation of this study.

Reviewer 3 Report

Comments and Suggestions for Authors

This paper assesses the recurrence-free survival (RFS) over 3 years as well as overall survival (OS) of melanoma, including acral and mucosal subtypes, treated with anti-PD-1 antibody in comparison to treatment with dabrafenib and trametinib. This is important since the incidence of mucosal and acral melanoma is higher in Asian cohorts, but these types were underrepresented in previous studies. I recommend publication with the following changes.

The Simple Summary should be removed, and the Abstract should be re-written for clarity. I suggest something like how I have re-written it below. However, if I have made any mistakes, of course those must be corrected.

“Abstract: Background Adjuvant therapy has improved the clinical prognosis for postoperative melanoma patients. However, the long-term efficacy of this therapy on the melanoma acral and mucosal subtypes has not been fully evaluated in previous trials. This study assessed the 3-year recurrence-free survival and overall survival of patients with melanoma, including acral and mucosal subtypes, treated with anti-PD-1 antibody (Ab) or with the combination of the BRAF and MEK inhibitors dabrafenib and trametinib. Methods We retrospectively analyzed both the 3-year recurrence-free survival (RFS) and overall survival (OS) of 120 patients treated with anti-PD-1 antibody (Ab), or with the combination of dabrafenib and trametinib. Results The overall median RFS was 18.4 months, with a range of 0.69 to 36 months. The 3-year RFS of the acral and mucosal types was 28.1% and 38.5%, respectively. Baseline tumor thickness (TT) and acral type were associated with the RFS in subgroup analysis. Moreover, we classified 104 acral and non-acral cutaneous patients into the anti-PD-1 Abs or dabrafenib plus trametinib combined therapies cohort in multiple analyses. The acral subtype and TT were detected as important prognostic factors. In the 3-year OS, only tumor ulceration was associated with the OS in both univariate and multiple analyses. There was no significant difference in baseline or treatment-related factors of mucosal type (P >0.05). Conclusion This study suggests that adjuvant therapy is more effective with non-acral cutaneous melanoma than either the acral or mucosal types at the 3-year RFS endpoint.”

Please fix or remove the link on p. 3. It did not work because it is broken across lines 114-115.

Figure 1 and 2 legends should have titles. For example,

Figure 1. Assessment of Recurrence-Free Survival in Post-Operative Melanoma

Figure 2. Assessment of Overall Survival in Post-Operative Melanoma

The sentence at 268-269 is confusing because it says “… both acral and mucosal types … are less effective…,” but it is the treatment that is less effective. I suggest writing it this way: “This study suggests that adjuvant therapy of both acral and mucosal melanoma types is less effective than adjuvant therapy of non-acral cutaneous melanoma at the 3-year RFS.”

Comments on the Quality of English Language

The quality of English is fine except for the changes suggested above.

Author Response

Thank you for your comment. We have corrected the abstract as you pointed out.

We have corrected the word as you pointed out.

Round 2

Reviewer 1 Report

Comments and Suggestions for Authors

1. Regarding studies of surgery with acral melanoma, please also comment on retrospective studies.  A literature search is waranted.  This reviewer found a few references quickly from reputable journals using google.  

2.  Placing all the variables into one regression model does not obviate the need for correction for multiple comparisons.  They are two different questions.  

Comments on the Quality of English Language

There are minor concerns now.  

Author Response

Comments and Suggestions for Authors

Query 1: Regarding studies of surgery with acral melanoma, please also comment on retrospective studies.  A literature search is waranted.  This reviewer found a few references quickly from reputable journals using google.  

Answer 1: We have added the references about retrospective studies of adjuvant therapy for acral melanoma additionally.

Query 2: Placing all the variables into one regression model does not obviate the need for correction for multiple comparisons.  They are two different questions.  

Answer 2: Multiple comparisons is a statistical technique in which an analysis of means between multiple groups is performed and, if there are multiple analyses with significant differences, the significance level is adjusted to 5% or less by post hoc HAC to clarify which analyses are truly significantly different. Although multiple comparisons of means are usually possible, there is no corresponding statistical method for survival.

Reviewer 2 Report

Comments and Suggestions for Authors

Can be published

Author Response

Thank you for your reviewing.